# Effect of Storage Time and Floral Origin on the Physicochemical Properties of Beeswax and the Possibility of Using It as a Phase Changing Material in the Thermal Storage Energy Technology

**DOI:** 10.3390/foods11233920

**Published:** 2022-12-05

**Authors:** Badria M. Al-Shehri, Thahabh Haddadi, Eman M. Alasmari, Hamed A. Ghramh, Khalid Ali Khan, Mohammed Elimam Ahamed Mohammed, Mohammed Sager Alotaibi, Mogbel Ahmed Abdalla El-Niweiri, Abdulrahman Hamdi Assiri, Maha M. Khayyat

**Affiliations:** 1Department of Chemistry, Faculty of Science, King Khalid University, Abha 61413, Saudi Arabia; 2Unit of Bee Research and Honey Production, King Khalid University, Abha 61413, Saudi Arabia; 3Research Centre for Advanced Materials Science (RCAMS), King Khalid University, Abha 61413, Saudi Arabia; 4King Abdul Aziz City for Science and Technology, Riyadh 11442, Saudi Arabia; 5Department of Biology, Faculty of Science, King Khalid University, Abha 61413, Saudi Arabia; 6Applied College, King Khalid University, Abha 61413, Saudi Arabia; 7Department of Bee Research, Environment and Natural Resources & Desertification Research Institute, National Center for Research, Khartoum P.O. Box 6096, Sudan; 8The Poison Control and Medical Forensic Chemistry Center, Asir Region, King Abdullah Road, Abha 62221, Saudi Arabia

**Keywords:** beeswax storage time, extraction method, color, intensity, phase changing material, thermal capacity

## Abstract

Beeswax is a natural product that is primarily produced by honey bees of the genus *Apis*. It has many uses in various kinds of industries, including pharmacy and medicine. This study investigated the effect of storage and floral origin on some physicochemical properties of four beeswax samples. The floral origin of the beeswax samples was determined microscopically and the investigated physical properties were the melting point, color, surface characteristics and thermal behavior. The studied chemical constituents were the acid value, ester value, saponification value and the ester/acid ratio. The FT-IR, SEM, EDX, XRD and TGF techniques were applied to meet the objectives of this study. The physical properties of the beeswax were affected by the storage period and floral origin. The melting point of the beeswax samples significantly increased with the increase in the storage time, from 61.5 ± 2.12 °C for the 3 month sample to 74.5 ± 3.54 °C for the 2 year stored sample (*p*-value = 0.027). The acid values of the 3 month, 6 month, 1 year and 2 years stored samples were 19.57 ± 0.95, 22.95 ± 1.91, 27 ± 1.91 and 34.42 ± 0.95 mgKOH/g, respectively. The increase in the acid value was significant (*p*-value = 0.002). The ester values of the studied beeswax samples significantly increased with the increase in storage time as follows: 46.57 ± 2.86 mgKOH/g for the 3 month stored sample, 66.14 ± 3.82 mgKOH/g for the 6 month stored sample, 89.77 ± 0.95 mgKOH/g for the one year stored sample and 97.19 ± 1.91 mgKOH/g for the 2 year stored sample (*p*-value ≤ 0.001). Similarly, the saponification value and the carbon percentages increased with the increase in storage time. Unlike the results of the chemical components, the oxygen percentage decreased with the increase in storage time as follows: 11.24% (3 month), 10.31% (6 month), 7.97% (one year) and 6.74% (two year). The storage and floral origin of beeswax significantly affected its physicochemical properties in a way that qualify it to act as a phase changing material in the thermal storage energy technology.

## 1. Introduction

Beeswax is a honey bee product that is mainly produced by honey bees of the genus Apis. It has many uses in various kinds of industries, in pharmacy and medicine. In the industrial sector, beeswax is used as hydrophobic insulator for numerous products such as electrical cables and circuits, leather and the shoe production industry, in inks and varnishes preparation and in the production of dried meat. Since the beeswax has healing and anti-inflammatory properties, it is used in ointments, creams and plasters. Medically, chewing gums containing beeswax are used for the purpose of inducing saliva and gastric juice secretion, removal of dental stones, reduction of nicotine in the breath of smokers and encapsulation of drugs [1,2,3]. Moreover, beeswax is one of the components of the edible coating of foods aiming to achieve long duration, maintaining their best physical properties and inhibiting microbial growth [4,5].

Beeswax is commercially produced by specialized beeswax factories across the world. Additionally, old bee combs or unrefined wax are provided by beekeepers; hence, the production procedures have a high impact on the quality of beeswax. Melting and chemical extraction are the two important procedures for extracting wax. Melting is the most used method. Wax can be melted using hot water, steam, electricity, or solar energy. Chemical solvent extraction is only possible in a laboratory [6].

Typically, beeswax as a raw material contains more than 300 chemical compounds, such as hydrocarbons, esters of fatty acids, and long chain alcohols, and falls under the category of organic non-paraffin phase change materials (PCMs) [7,8,9]. The constituents of beeswax are classified into major and minor components. The major components are those constituting more than 1% and they include hydrocarbons, free acids, free alcohols, esters (monoesters, diesters, hydroxyl mono and polyesters, and acid esters) and unidentified compounds [10,11]. Beeswax is an inert material with high plasticity at a relatively low temperature (around 32 °C). By contrast, most plant waxes are much harder at this temperature and are of crystalline structure; upon heating, the physical properties of wax change. At 30–35 °C it becomes plastic, at 46–47 °C the structure of a hard body is destroyed, and between 60 to 70 °C it begins to melt. Heating to 95–105 °C leads to the formation of surface foam, while at 140 °C the volatile fractions begin to evaporate. After cooling down, beeswax shrinks by about 10% at 120 °C for at least 30 min, causing an increase in hardness due to the removal of the remaining water [12]

The quality of wax is strongly affected by some factors such as production procedure, the melting point, the kind of honey bees, the geographical condition, the harvest method, and the age of the beeswax [13]. For example, the physical properties such as color, odor, stiffness, elasticity, crystallization and hardness are affected by the age of the wax [14,15].

In recent times, FTIR-ATR spectroscopy might be used to identify changes in the physical and chemical properties of biomaterials such as beeswax. The relative differences in the absorption intensity bands spectra of FTIR of different ages of beeswax samples can be identified and calculated through comparison [16].

In this study, the authors investigated the effect of the storage time of beeswax on some of effective physio-chemical properties, such as the color of samples and the chemical contents of the thermal capacity feature. The different characteristics of technologies were applied to clarify the studied samples. Finally, the authors found that the recent results might contribute to the institution of a new library for future studies of different advanced industrial and cosmetics applications.

## 2. Materials and Methods

### 2.1. Collection, Location, Preparation of Samples

All crude beeswax combs samples were collected from hives of *Apis mellifera jemenitica* honeybee farms at King Khalid University, Saudi Arabia from Jan 2020 to October 2021. Four samples (denoted 3-M, 6-M, 1-Y, 2-Y) were harvested without being reused and stored for 3 months, 6 months, 1 year, and 2 years, respectively (Table 1) [17].

The beeswax samples were melted into an aqueous solution at 60–65 °C for one hour, filtrated 3 to 5 times using a stainless steel filter, cooled and collected from the top of the filtrate, and finally, stored in cold and dark conditions.

### 2.2. Colour Determination

The color of the beeswax samples was determined by comparing its color to the Pfund scale for honey [18] as presented in (Table 1) and (Figure 1).

### 2.3. Floral Origin of Honey Stored in the Beeswax

To investigate the floral origin of the honey stored in the beeswax, the wax samples were melted in hot water and the water was scanned under a microscope for the presence of pollens [19].

### 2.4. Characterization of Beeswax Samples

#### 2.4.1. X-ray Diffraction (XRD)

Four Samples of beeswax were characterized by XRD to identify the crystalline phases. Wide-angle XRD analysis was performed on the EQUINOX 1000 X-ray Diffractometer (Thermo Scientific, Massachusetts, USA) operating at conditions of 40 KV and 30 mA, with a CuKα radiation. The diffracted intensities corresponding to the 2θ angle were collected from 4 to 100° by means of a lynx eye detector [20].

#### 2.4.2. Fourier Transform Infrared Spectroscopy (FTIR)

The chemical features of the samples were determined via Fourier transform infrared spectrometer (FTIR, Hopkinton, MA, USA), with high resolutions of up to 0.15 cm^−1^ in the range from 450 to 4500 cm^−1^. The beeswax was carried out following the method of Svečnjak et al. (2015) [16].

#### 2.4.3. Scanning Electron Microscopy (SEM) and Energy-Dispersive X-ray Analyzer (EDX)

The surfaces of beeswax samples were studied using a scanning electron microscope (JEOL Ltd., JSM-7100F, Tokyo, Japan) with an accelerating voltage of 5–20 kV. All samples were coated with a thin gold film (~5 nm) to avoid the surface charge and to enhance the emission beams of secondary electrons, thus ensuring that the specimen conducts uniformly and produces a uniform surface for imaging and analysis with high resolution. Additionally, beeswax samples were placed on a cylindrical aluminum (sample holder) and fixed with carbon tape. Moreover, an Energy-Dispersive X-ray Analyzer (EDX) is used to investigate the elements’ quantitative composition. EDX investigates the energy and intensity distribution of X-rays generated by electron beam excitation on the material’s surface, allowing for a highly accurate estimation of the elemental composition throughout the specified area covered by the electron beams [20].

#### 2.4.4. Thermogravimetric Analyses (TGA)

Thermogravimetric Analysis (TGA) is immensely attractive for recognizing solid fuel degradation and its release of energy. This technique can track the thermal degradation arrangements of fresh biomaterials. The understanding of beeswax’s thermal behavior is critical in identifying the thermal degradation (or stability) of the substance, the conversion of the substance, and the product formation. The beeswax samples were placed in alumina crucibles under a 100 mL/min flow rate of nitrogen and a 10 °C/min heating rate, using a thermo-balance Discovery SDT 650 (TA Instruments, New Castle, DE, USA) [21].

#### 2.4.5. Determination of the Acid Value

The amount of KOH in mg used to neutralize one gram of beeswax is known as the acid value. The experiment was performed according to the testing methods of the American Wax Importers and Refiners. The following steps were followed: approximately 3 g of the beeswax sample were weighed to an accuracy of 1 mg and transferred to a 250 mL flask. Then, 50 mL of neutralized isopropanol and toluene 5:4 and 5 drops of phenolphthalein indicator were added and dissolved under a reflux condenser. When completely dissolved, the mixture of the beeswax was titrated with 0.5 N Methanolic (95%) potassium hydroxide to a permanent pink color (does not disappear after very gently boiling for one and a half minute) [22].

The acid value was calculated using the following formula:(1)$$\mathrm{Acid}\;\mathrm{value}\;\left(\mathrm{mgKOH}/g\right)=\frac{\mathrm{AN}\times 56.1}{C}$$
where:A—Milliliters of KOH solution required for the titration of the sampleN—Normality of the KOH solution.C—Grams of the samples used.56.1—Molecular mass of KOH

#### 2.4.6. Measurement of the Ester Value

The ester value is defined as the amount of KOH in mg needed to neutralize ester linked acids in one gram of beeswax. To the solution resulting from the determination of the acid value, exactly 15.0 mL of 0.5 N Methanolic (95%) potassium hydroxide was added. In another 250 mL flask (blank), 50 mL of neutralized isopropanol and Toulon 5:4 exactly, 15.0 mL of 0.5 N Methanolic (95%) potassium hydroxide and 5 drops of phenolphthalein test solution were added. The beeswax sample and the blank were subjected to reflux for 4 h, and the excess KOH was back titrated with 0.5 N Aqueous Hydrochloric Acid until the reddish tinge disappeared. Then the two solutions were boiled vigorously for several minutes to confirm the disappearance of the reddish tinge (American Wax Importers and Refiners) [23].The ester value was calculated using the below equation:(2)$$\mathrm{Ester}\;\mathrm{value}\;\left(\mathrm{mgKOH}/g\right)=\frac{\left(B-A\right)N\times 56.1}{C\;}$$
where:A—Milliliters of Hydrochloric Acid required for the titration of the sample.B—Milliliters of Hydrochloric Acid required for titration of the blank.N—Normality of the Hydrochloric Acid.56.1—Molar mass of KOHC—Grams of the sample used

#### 2.4.7. Evaluation of the Saponification Value

The saponification value is the amount of KOH in mg that is required for the neutralization of one gram of beeswax after alkaline hydrolysis (saponification). The saponification value was calculated following the below equation
(3)$$\mathrm{Saponification}\;\mathrm{value}\;\left(\mathrm{mg}\frac{\mathrm{KOH}}{g}\right)=\mathrm{Acid}\;\mathrm{value}+\;\mathrm{Ester}\;\mathrm{value}\;$$

#### 2.4.8. Ester/Acid Ratio

The ratio number is calculated by dividing the ester value by the acid value.

#### 2.4.9. Melting Point

The capillary tube method was used to determine the melting point of the beeswax samples using the Ry-2 melting point tester (RY-2 Melting Point Tester, SINOPHAM, Shanghai, China). The melting point of each sample was determined three times and the mean of the three readings was considered as the melting point of the sample [24].

### 2.5. Statistical Analysis

The Analysis of Variance test (ANOVA) of the Statistical Package for Social Sciences (SPSS-v20) was used to compare the mean values of the studied chemical parameter and the significant difference was set at the level of 5%.

## 3. Results and Discussion

### 3.1. The Floral Origins of the Beeswax Samples

The microscopic analysis of the beeswax showed that the samples were of four origins. Sample (2-Y) was of *Eucalyptus* origin (Figure 2) and this is proven because the bee farm is located near some Eucalyptus trees. Sample (1-Y) originated from *Cactus opuntia* (Figure 2), however, *Cactus opuntia* is a wild plant spread throughout King Khalid University bee farm. Samples (6-M) and (3-M) are of polyfloral origins. The 6-M sample contained pollens from *Ziziphus, Cactus* and *Acacia,* while the 3-M sample contained *Acacia* and *Atriplex halimus* pollens. All the pollen plants are available around the bee farm of KKU.

### 3.2. Effect of Storage Time on the Beeswax Samples Color

The color of the four samples is listed in Table 1, where the fresh sample was water white and changed with the increase in the storage time of the beeswax to white, extra white, and to an extra light amber color. Generally, beeswax contains an amount of pollen, propolis, and larval excrements in addition to other compounds; however, the amount of these components varies with the age of beeswax, and the increased storage time of the beeswax comb causes its color to turn darker. This change in the degree of color is considered to be evidence of the relative number of pigments of these colorant components, such as propolis and pollen. The darker color means an increase in these pigments, with increasing negative effects on the quality of beeswax [25,26]. The factors that are involved in the darkening of beeswax include high temperature heating, and heating in containers that contain aluminum, steel, copper or zinc [27]. Other factors that are involved in the determination of the beeswax color and physical properties include the floral origin, environmental pollution and honeybee species [28,29,30]

### 3.3. Crystalline Degree of Samples

The XRD patters give useful results regarding the crystallinity degree. Figure 3 illustrates the XRD patterns of beeswax samples at different intervals of storage time (three months, six months, one year, and two years) and shows the diffractograms. By a general comparison of four samples, there are two zones of multisignals in each pattern. The first zone in the 3-M sample was from 17.3° to 33° and the second zone was from 44° to 51°. The increase in storage time causes a shift in the peaks to the low angle by almost 5 degrees in each zone. However, the 6-M sample comes in the range 12–20° and 27–36°. The 1-Y sample showed the peaks in two ranges: 9.7–19.5° and 21° to 33°. Meanwhile, the 2-Y sample was in the range 8–19.5° for first the zone and 20–44° for second zone. Furthermore, the intensity of multi-signals is variable. However, the intensity increased in the first zone from 75° to almost 97°, with the storage time increased from 3 months to 2 years, indicating an increase in crystalline type as reported in [28,31]. In contrast, the intensity clearly decreased in the second zone from 124° to almost 56° with an increase in the storage time of the sample. Hence, the crystallinity of older beeswax, 2-Y and 1-Y, is better than the younger beeswax, 6-M and 3-M, where intensities are considered as follows: 2-Y > 1-Y > 6-M ≈ 3-M. These results match other results regarding ester and acid values.

### 3.4. FTIR Analysis Results

The recognizable infrared absorption bands of four prepared samples are presented in Figure 4. The absorption band’s positions, which are visible at 2960, 2913, 2842, 1739, 1464, 1171, and 714 cm^−1,^ are almost identical [10,14]. However, all spectra displayed a high number of bands in the region from 1750 cm^−1^ to 700 cm^−1^, which is considered a fingerprint of beeswax [32]. Furthermore, the asymmetric stretching vibrations of the CH_3_ group appeared at 2961 cm^−1^, while the CH_2_ group (stretching, binding, and rock) appeared at 2913, 2842, 1464, and 714 cm^−1^ [33]. The sharp characteristic bands of monoesters are recorded at 1739 cm^−1^ and 1171 cm^−1^, indicating the stretching and binding vibrations of the carbonyl group of monoesters [34]. More importantly, the characteristic stretching vibration of free fatty acid at 1714 cm^−1^ was recorded as a slight hump compared with a sharp band of esters [10].

Typically, the beeswax samples consist mainly of monoesters, containing more than 35% of higher fatty acids, in addition to some other hydrocarbons and free acids. Once of the characteristic methods to describe the storage time of beeswax is to calculate the ratio of ester values to fatty acids. However, many factors are affecting this ratio, such as the high boiling point,, the storage time, and the age of the beeswax.

### 3.5. Thermogravimetric Analysis (TGA)

Thermogravimetric analysis (TGAs) is an effective method for studying materials’ composition and thermal stability. Figure 5 demonstrates the TGA results conducted on BW samples at the different storage times of two years, one year, six months, and three months for 2-Y, 1-Y, 6-M, and 3-M, respectively. As shown (Figure 5d), the 2-Y (two years aged) exhibited weight loss from approximately 150 to 390 °C, and the green curve of the derivatives revealed multiple weight losses at 259.8 and 350.5 °C until the complete decomposition of beeswax at 380 °C. For one-year-old 1-Y, weight loss occurred between 130 and 380 °C, with significant losses at 245.8 and 340.4 °C, as indicated by the blue derivatives curve; weight loss continued until complete dissociation at 380 °C. For 6-M (6 months) and 3-M (3 months), nearly identical weight loss curves were observed between 125 and 370 °C. In addition, the derivatives curves demonstrated multiple losses at temperatures (241.5 and 340.1 °C for 6-M, and 241.2 and 338.8 °C for 3-M) before they completely decomposed at 370 °C. All beeswaxes of different ages exhibited multiple weight loss results due to the multi-component material structure. Additionally, the high thermal stability is shown by the fact that the beeswaxes were not entirely decomposed until the temperature approached ~380 °C. Furthermore, as demonstrated by the weight loss derivative curves, the elder 2-Y (2 years) requires a higher temperature to start weight loss, at 150 °C, and completely decomposed at 390 °C, compared to the younger 6-M (6 months) and 3-M (3 months) at 125 to 370 °C. Recently, Schuman studied different yellow and white beeswaxes and discovered, using TGA analysis, that weight loss began at 135 °C, and decomposition was completed at 430 °C, which is very close to our results of weight losses for 1-Y (130–380 °C) and 6-M and 3-M (125–370 °C). The TGA technique is well known to detect the thermal stability of materials and their volatile chemical constituents [35]. Examples of volatile compounds in beeswax include alkanes (2,4-dimethyl-heptane, 4-methyl-octane, nonane, dodecane, nonadecane and tetradecane), aldehydes (benzaldehyde, decanal, furfural, nonanal and octanal), alcohols (1-heptanol, 2-ethyl-1-hexanol, 1-octanol, 1-nonanol, benzyl alcohol and phenylethyl alcohol) and terpenes (α-pinene) [36].

### 3.6. SEM and EDX Characterizations

Scanning electron microscopy (SEM) is the most popular and powerful technology for surface analysis with high resolution. SEM, coupled with X-ray analysis, is considered a relatively quick and essentially nondestructive technique for surface examination. Figure 6 depicts SEM images of BW surfaces at the ages of two years, one year, six months, and three months for 2-Y, 1-Y, 6-M and 3-M, respectively. SEM images of beeswax surfaces show superimposed plates, as observed more in 1-Y and 6-M with some air pockets, due to the heterogeneous and amorphous nature of the polymers forming the beeswax. To determine the composition of beeswax, each sample was analyzed using EDX, as illustrated in Figure 7. The EDX findings indicate that the primary component of beeswax is carbon and oxygen. In addition, the carbon weight increases progressively with the increase in storage time from 88% to 93%, in contrast to the oxygen weight, which decreased with the increase in storage time from 11.44% to 6.74%, as listed in Table 2 [37].

### 3.7. Effect of Storage Time on the Chemical Compounds

#### 3.7.1. Melting Point

The mean melting point of the beeswax increased with the increase in storage time, from 61.5 °C at the three month storage, to 74.5 °C at the two years storage period (Table 2). The beeswax stored for two years had a significantly increased melting point, compared to the six month sample (*p*-value = 0.041), and compared to the three month beeswax sample (*p*-value = 0.007). Table 3 shows that the melting point of the beeswax depends on the honeybee species, and it ranges from 51.9 °C to 85.3 °C [35].

#### 3.7.2. Acid Value

The storage time gradually decreased the acid value of the beeswax samples. The three month stored beeswax displayed a significantly higher acid value than the six month stored sample (*p*-value = 0.008), one year stored sample (*p*-value = 0.002) and the two years stored sample (*p*-value = 0.001) (Table 2 and Table 3). The acid value of beeswax is reported to be within the range of (17–24 mgKOH/g) [25]. The acid values of the three and six month stored beeswaxes are higher (27 and 34.42  mgKOH/g) when compared to the literature, and this may be due to the floral source and climate differences.

#### 3.7.3. Ester Value

The ester value of the beeswax samples significantly increased with the increase in storage time (Table 2 and Table 3). The ester values of the beeswax stored for three and four months of this study are less than the previously reported values [25], which may be attributed to the floral origin of the beeswax. The range of the ester value of the beeswax was reported to be (70–90 mgKOH/g) [25].

#### 3.7.4. Saponification Value

The saponification value (number) increased as the storage time increased (Table 2). The three month beeswax sample was significantly lower than all of the other beeswax samples (Table 3). Bogdanov et al. (2016) mentioned that the saponification value of the beeswax ranges from 87 to 102 mgKOH/g [25]. The saponification value of the beeswaxes of this study is wider than the previous studies (81–116.76 mgKOH/g).

#### 3.7.5. Ester/Acid Ratio

The ester/acid ration significantly increased with the increase in storage time (Table 2 and Table 3). The range of the ester/acid ration of the beeswax is (3.3–4.3) [25]. This study reported a wider range of the ester/acid ratio of the beeswax (1.35–4.97). The difference between the range of the ester/acid ratio may be due to the effects of the floral origin and storage.

The storage significantly affected all of the studied parameters by different levels, depending on the storage duration.

## 4. Conclusions

The produced results proved that the physicochemical properties of the beeswax were strongly affected by the time of storage and the floral origin, which in turn affected the thermal stability and mechanical properties of beeswax. The results of this study qualify the beeswax to be used as a promising phase changing material in the thermal storage energy technology.

## Figures and Tables

**Figure 1 foods-11-03920-f001:**
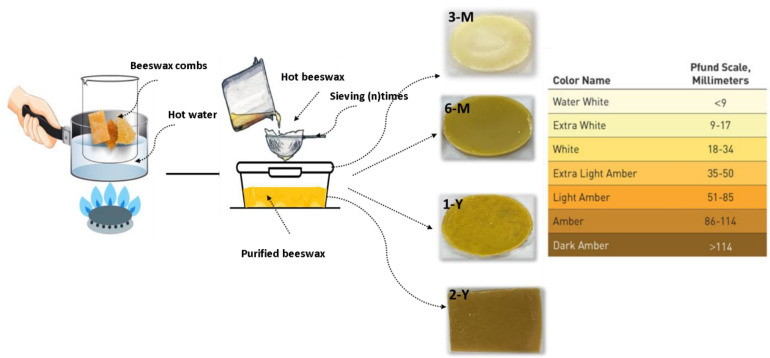
The melting extraction method of four BW samples with different storage times [17,18].

**Figure 2 foods-11-03920-f002:**
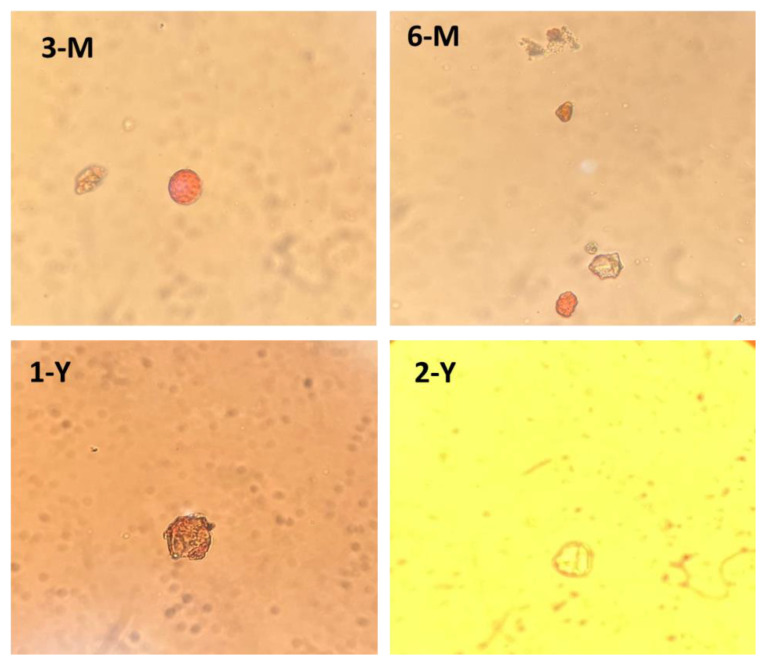
The pollens of the studied beeswax samples. 3-M and 6-M were of polyfloral origins. Meanwhile, 1-Y originated from Cactus opuntia and 2-Y was of Eucalyptus origin.

**Figure 3 foods-11-03920-f003:**
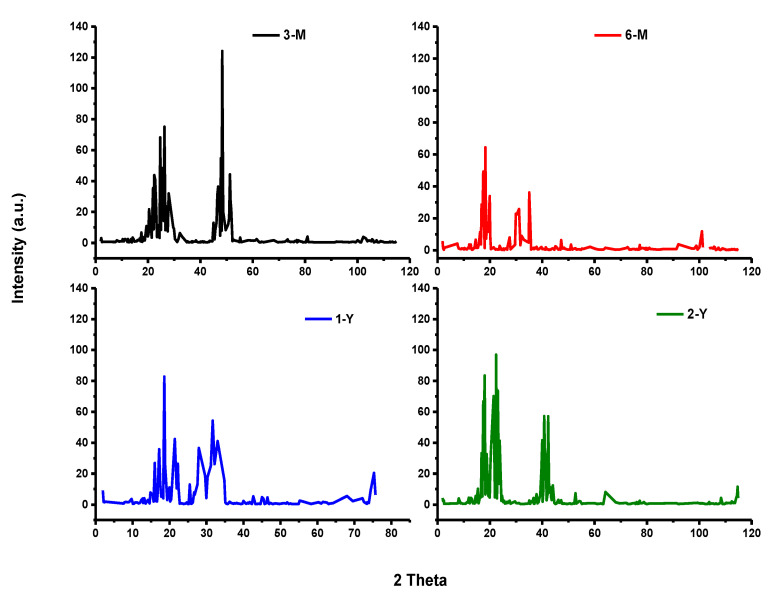
X-ray diffractograms of the beeswax samples with different storage times.

**Figure 4 foods-11-03920-f004:**
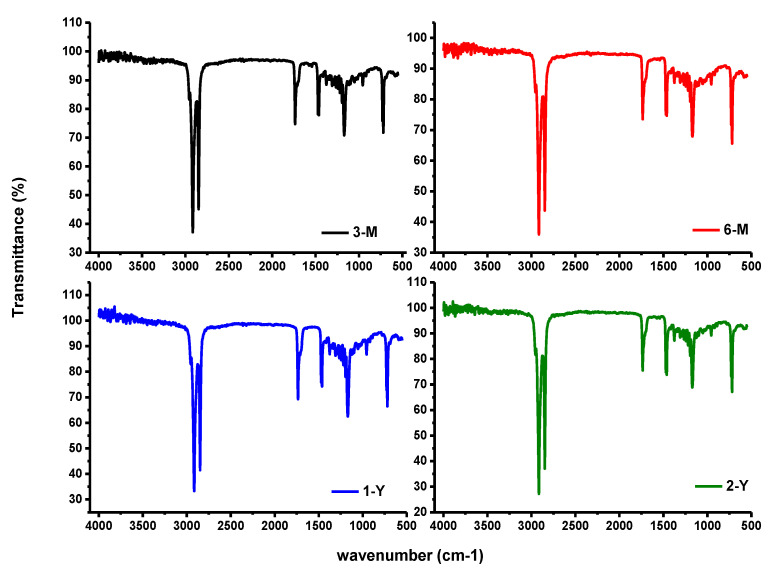
Fourier Transform Infrared Spectra (FTIR) of four BW samples with different storage times.

**Figure 5 foods-11-03920-f005:**
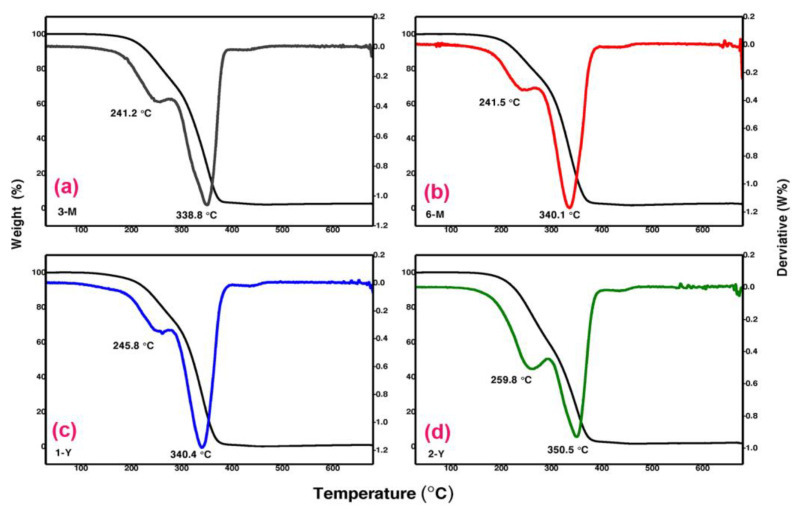
Thermogravimetric curves and their derivatives from room temperature to 700 °C for BW samples at different storage times: (**a**) 3-M aged three months, (**b**) 6-M aged six months, (**c**) 1-Y aged one year, and (**d**) 2-Y aged two years. The thin black lines represent weight loss curves, while the colored lines represent weight loss derivatives.

**Figure 6 foods-11-03920-f006:**
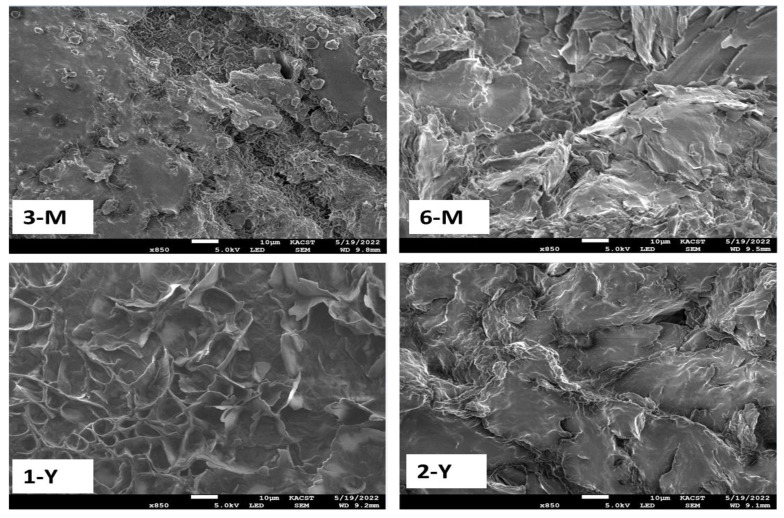
SEM micrographs of four BW samples with different storage times.

**Figure 7 foods-11-03920-f007:**
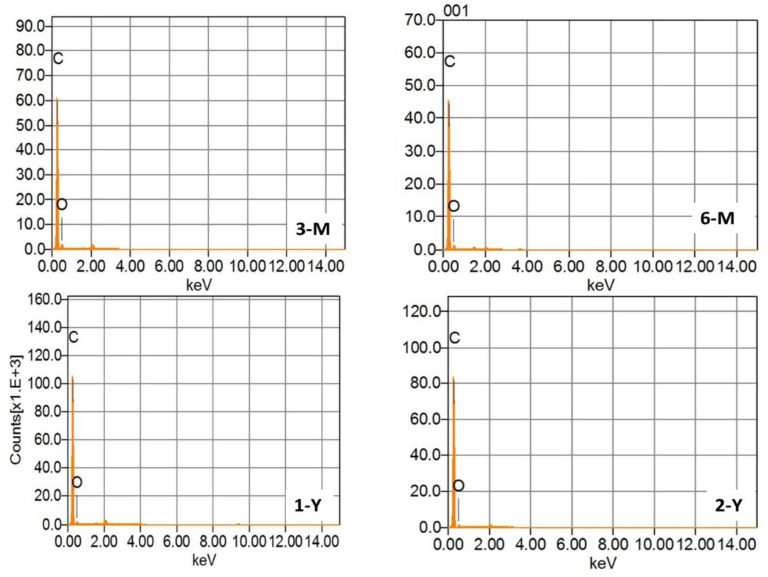
Energy dispersive X-ray analysis (EDX) spectrums of four BW samples with different storage times.

**Table 1 foods-11-03920-t001:** General information about the studied beeswax samples.

The Sample Label	Harvest Date	Beeswax Color	Storage Time
2-Y	January 2020	Extra light amber	2 Years
1-Y	January 2021	Extra white	1 Year
6-M	July 2021	White	6 months
3-M	October 2021	Water white	3 months

**Table 2 foods-11-03920-t002:** Chemical constituents of the different beeswax samples.

The Sample Label	2-Y	1-Y	6-M	3-M
Melting point	74.5 ± 3.54 °C	70.5 ± 0.71 °C	67 ± 2.83 °C	61.5 ± 2.12 °C
Acid value mgKOH/g	19.57 ± 0.95	22.95 ± 1.91	27 ± 1.91	34.42 ± 0.95
Ester value mgKOH/g	97.19 ± 1.91	89.77 ± 0.95	66.14 ± 3.82	46.57 ± 2.86
Saponification value mgKOH/g	116.76 ± 0.95	112.71 ± 0.95	93.14 ± 1.91	81 ± 3.82
Ester/Acid	4.97 ± 0.34	3.93 ± 0.37	2.46 ± 0.42	1.35 ± 0.05
Carbon %	93.26	92.03	89.69	88.76
Oxygen %	6.74	7.97	10.31	11.24

* The carbon and oxygen percentages were calculated for one sample, and for that reason there is no average value ± standard deviation.

**Table 3 foods-11-03920-t003:** Statistical differences between the mean values of the studied parameters.

Parameter			*p*-Value *	General *p*-Value
Melting point	3 month beeswax	6 month beeswax	0.095	0.027
		One year beeswax	0.023	
		Two years beeswax	0.007	
	6 month beeswax	One year beeswax	0.238	
		Two years beeswax	0.041	
	One year beeswax	Two years beeswax	0.188	
Acid value	3 month beeswax	6 month beeswax	0.008	0.002
		One year beeswax	0.002	
		Two years beeswax	0.001	
	6 month beeswax	One year beeswax	0.055	
		Two years beeswax	0.008	
	One year beeswax	Two years beeswax	0.089	
Ester value	3 month beeswax	6 month beeswax	0.002	≤0.001
		One year beeswax	≤0.001	
		Two years beeswax	≤0.001	
	6 month beeswax	One year beeswax	0.001	
		Two years beeswax	≤0.001	
	One year beeswax	Two years beeswax	0.047	
Saponification number	3 month beeswax	6 month beeswax	0.006	≤0.001
		One year beeswax	≤0.001	
		Two years beeswax	≤0.001	
	6 month beeswax	One year beeswax	0.001	
		Two years beeswax	≤0.001	
	One year beeswax	Two years beeswax	0.145	
Ester/Acid ratio	3 month beeswax	6 month beeswax	0.002	0.001
		One year beeswax	0.001	
		Two years beeswax	≤0.001	
	6 month beeswax	One year beeswax	0.008	
		Two years beeswax	0.001	
	One year beeswax	Two years beeswax	0.024	

## Data Availability

The data is included in this article.

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
