# Peer review of "Effect of Storage Time and Floral Origin on the Physicochemical Properties of Beeswax and the Possibility of Using It as a Phase Changing Material in the Thermal Storage Energy Technology"

_foods, 2022, doi:10.3390/foods11233920_

Round 1

Reviewer 1 Report

The manuscript "Characterization of beeswax stored at different time intervals 2 and its TGA analysis for possible application as a thermal energy storage material" shows results on the analysis of 4 beeswax samples of different production date.

Nowadays there are few works on the characterization of beeswax which is one of the important products produced in the beehive after honey.

The work provides interesting background information on the wax components according to FTIR and TGA results.

It is important to note that only 4 beewax samples were analyzed and the geographical origin with botanical reference is not specified. It would be good to add a brief description of the flora surrounding the beehives and the type of honey produced in them, since this is one of the factors that conditions the color and appearance of the wax and is mentioned by the authors in the introduction.

It is also important to keep in mind that the age of the wax is also related to the time of use in the beehive. Beekeepers frequently reuse the beewax and in this process the matrix undergoes the changes that the authors describe in the study. If a wax is produced at a certain time and stored without being used, it will probably maintain its properties over time and will behave similarly years later when compared to a newly produced beewax.

It would be interesting to compare and mention the possible applications that this study may have in other matrices of similar origin and composition (e.g. oils and fats). Nowadays, the search for products of natural origin that possess properties and whose uses have diversified over time could support the scope of the results of this study. 

Specifically, all figures should be increased in size to facilitate reading and analysis of the results.

Author Response

REVIEWER-1

Comments and Suggestions for Authors

The manuscript "Characterization of beeswax stored at different time intervals  and its TGA analysis for possible application as a thermal energy storage material" shows results on the analysis of beeswax samples of different production date.

Nowadays there are few works on the characterization of beeswax which is one of the important products produced in the beehive after honey.

The work provides interesting background information on the wax components according to FTIR and TGA results.

  1. It is important to note that only beewax samples were analyzed and the geographical origin with botanical reference is not specified. It would be good to add a brief description of the flora surrounding the beehives and the type of honey produced in them, since this is one of the factors that conditions the color and appearance of the wax and is mentioned by the authors in the introduction.

Pollens deposited in the beeswax samples were investigated microscopically and the floral sources were determined. A new figure of the beeswax pollens is added. The objectives, methods, results and discussion were modified accordingly.  

  1. It is also important to keep in mind that the age of the wax is also related to the time of use in the beehive. Beekeepers frequently reuse the beewax and in this process the matrix undergoes the changes that the authors describe in the study. If a wax is produced at a certain time and stored without being used, it will probably maintain its properties over time and will behave similarly years later when compared to a newly produced beewax.

We have mentioned in the description of the beeswax samples that the beeswax samples are harvested without being reused.

  1. It would be interesting to compare and mention the possible applications that this study may have in other matrices of similar origin and composition (e.g. oils and fats). Nowadays, the search for products of natural origin that possess properties and whose uses have diversified over time could support the scope of the results of this study. 

We have carried out two new experiments and determined the content of the free fatty acids and esters.

  1. Specifically, all figures should be increased in size to facilitate reading and analysis of the results.

The figures are loaded in the online system separately to facilitate the review process.

Reviewer 2 Report

Review of MS: foods-1987000

Title: Characterization of beeswax stored at different time intervals and its TGA analysis for possible application as a thermal energy storage material.

By: AL-Shehri et al.

General Comment.

This work is on a relatively unknown aspect of beeswax, and although it is centered in few studies, it is interesting. However, I consider that this study must be complemented with at least a study on the chemical characterization of beeswax of different ages (changes in the main hydrocarbons, free fatty acids, and esters) or a study of rheological properties of beeswax stored for different times are needed.

Specific Comments

Abstract. Although there are some numerical figures about the work carried out, I consider necessary to include some more numerical data about the results of this research. Please correct.

L. 88-89. More data are needed to specify the species of honey bees used to produce the beeswax, and the flowers from which the bees were fed.  

L. 103-104. The authors need to mention the time used to heat the beeswax in the water bath and the type of filter used to purify the beeswax. Please clarify.

109-111. The manufacturer name of the FTIR equipment is required, as well as the superscripts written here.

L. 118. I believe the correct term used to name the TGA analyzer is “thermal analyzer”. Please correct.

L. 134. The method for color determination is missing. Please correct.

L. 237. Other reports have found that beeswax contains 42.8-68.2 % of C, but in this work three samples showed 100% of C. Please clarify.

Author Response

REVIEWER-2

Comments and Suggestions for Authors

Review of MS: foods-1987000

Title: Characterization of beeswax stored at different time intervals and its TGA analysis for possible application as a thermal energy storage material.

By: AL-Shehri et al.

General Comment.

This work is on a relatively unknown aspect of beeswax, and although it is centered in few studies, it is interesting. However, I consider that this study must be complemented with at least a study on the chemical characterization of beeswax of different ages (changes in the main hydrocarbons, free fatty acids, and esters) or a study of rheological properties of beeswax stored for different times are needed.

We have carried out two new experiments and determined the content of the free fatty acids and esters. Regarding the hydrocarbons, we were not able to analyze them because the GC. MS is not functioning, we are very sorry about that.

 Specific Comments

  1. Although there are some numerical figures about the work carried out, I consider necessary to include some more numerical data about the results of this research. Please correct.

All the abstract is rephrased including your suggestion.

  1. 88-89. More data are needed to specify the species of honey bees used to produce the beeswax, and the flowers from which the bees were fed.

We have added the honeybee species (Apis mellifera jemenitica

  1. 103-104. The authors need to mention the time used to heat the beeswax in the water bath and the type of filter used to purify the beeswax. Please clarify.

Done in the experimental section (2.1. Description and preparation of the beeswax samples).

  1. 109-111. The manufacturer name of the FTIR equipment is required, as well as the superscripts written here.

Modified.

  1. 118. I believe the correct term used to name the TGA analyzer is “thermal analyzer”. Please correct.

Done.

  1. 134. The method for color determination is missing. Please correct.

Done (2.1. Description and preparation of the beeswax samples).

  1. 237. Other reports have found that beeswax contains 42.8-68.2 % of C, but in this work three samples showed 100% of C. Please clarify.

Clarified

Reviewer 3 Report

REVIEW COMMENTS

1. Please delete lines 23-24 in to Abstract: > Wax can be melted using hot water, steam, electricity, or solar energy<, because authors use hot water<

2. The lines 134-136 > The color of the four samples is listed in table 1, where the fresh sample was 135

white and changed with the age of beeswax samples to yellow, dark yellow, and brown 136

color< has not are correct correspondent. Table 1 content other data. Please change and addition data.

3. The methods (all methods) must have bibliographic references<

4. The changes in chemical composition over time were not explained, Lines >223-237.

5. The research results were not statistically interpreted, no correlations were made between the analyzed variables. It is recommended to use a mathematical modeling.

Author Response

REVIEWER-3

Comments and Suggestions for Authors

REVIEW COMMENTS

  1. Please delete lines 23-24 in to Abstract: > Wax can be melted using hot water, steam, electricity, or solar energy<, because authors use hot water<

The abstract is rewritten as requested by all the reviewers

  1. The lines 134-136 > The color of the four samples is listed in table 1, where the fresh sample was 135

changed

white and changed with the age of beeswax samples to yellow, dark yellow, and brown 136

changed according the pfund scale

color< has not are correct correspondent. Table 1 content other data. Please change and addition data.

changed

  1. The methods (all methods) must have bibliographic references<

done

  1. The changes in chemical composition over time were not explained, Lines >223-237.

done

  1. The research results were not statistically interpreted, no correlations were made between the analyzed variables. It is recommended to use a mathematical modeling.

done

Round 2

Reviewer 2 Report

Review of MS 1987000 V.2

Title: Characterization of beeswax stored at different time intervals and its TGA analysis for possible application as a thermal energy storage material.

By AL-Shehri et al.

General comment

This report was appropriately adjusted, adding what the reviewers asked. I consider that English language still requires some more clarification. There are other minor points that require clarification and are written below.

Specific comments

L. 33-35. I consider that numerical values of the chemical evaluation of the different waxes at different storage times should be displayed, instead of the p-values. Please correct.

L. 147, 160. The methanolic ethanol solution requires the percentage methanol used.

I consider that equations need to be numbered. Please correct

L. 184. This sentence is not clear and needs clarification, and in addition the equipment used requires manufacturer, city and country of fabrication. Please correct.

L. 190. I believe that the version of the SPSS software is not correct. Please clarify.

Table 3. The average values of the physicochemical properties should be written in the table. Please correct.

Author Response

REVIEWERS COMMENTS

REVIEWER-2

GENERAL COMMENTS

This report was appropriately adjusted, adding what the reviewers asked. I consider that English language still requires some more clarification. There are other minor points that require clarification and are written below.

Answer: English language improved in revised version.

SPECIFIC COMMENTS

  1. 33-35. I consider that numerical values of the chemical evaluation of the different waxes at different storage times should be displayed, instead of the p-values. Please correct.

Answer: Numerical values of the chemical evaluation of the different waxes at different storage times are displayed in revised text.

  1. 147, 160. The methanolic ethanol solution requires the percentage methanol used.

Answer: Done according to suggestion.

I consider that equations need to be numbered. Please correct

Answer: All equations are numbered now according to suggestion.

  1. 184. This sentence is not clear and needs clarification, and in addition the equipment used requires manufacturer, city and country of fabrication. Please correct.

Answer: Corrected accordingly.

  1. 190. I believe that the version of the SPSS software is not correct. Please clarify.

Answer: Corrected accordingly.

Table 3. The average values of the physicochemical properties should be written in the table. Please correct.

Answer: The carbon and oxygen percentages were calculated for one sample for that reason there is no average value± standard deviation